# A Dual Strategy—In Vitro and In Silico—To Evaluate Human Antitetanus mAbs Addressing Their Potential Protective Action on TeNT Endocytosis in Primary Rat Neuronal Cells

**DOI:** 10.3390/ijms25115788

**Published:** 2024-05-26

**Authors:** Cauã Pacheco Lima, Gabriela Massaro Barreiros, Adriele Silva Alves Oliveira, Marcelo Medina de Souza, Tania Maria Manieri, Ana Maria Moro

**Affiliations:** 1Laboratory of Biopharmaceuticals, Butantan Institute, Sao Paulo 05503-900, Brazil; caua.lima@usp.br (C.P.L.); gabriela.barreiros@usp.br (G.M.B.); adriele.soliveira@butantan.gov.br (A.S.A.O.); 2Interunits Graduate Program in Biotechnology, University of Sao Paulo, Sao Paulo 05508-270, Brazil; 3CENTD—Centre of Excellence in New Target Discovery, Butantan Institute, São Paulo 05503-900, Brazil; marcelo.souza@butantan.gov.br; 4CeRDI—Center for Research and Development in Immunobiologicals, Butantan Institute, São Paulo 05503-900, Brazil

**Keywords:** ganglioside, fragment C of tetanus toxin, synaptic vesicle 2, rat primary spinal cord cells, epitope prediction, conformational epitopes, molecular docking

## Abstract

Tetanus disease, caused by *C. tetani*, starts with wounds or mucous layer contact. Prevented by vaccination, the lack of booster shots throughout life requires prophylactic treatment in case of accidents. The incidence of tetanus is high in underdeveloped countries, requiring the administration of antitetanus antibodies, usually derived from immunized horses or humans. Heterologous sera represent risks such as serum sickness. Human sera can carry unknown viruses. In the search for human monoclonal antibodies (mAbs) against TeNT (Tetanus Neurotoxin), we previously identified a panel of mAbs derived from B-cell sorting, selecting two nonrelated ones that binded to the C-terminal domain of TeNT (HCR/T), inhibiting its interaction with the cellular receptor ganglioside GT1b. Here, we present the results of cellular assays and molecular docking tools. TeNT internalization in neurons is prevented by more than 50% in neonatal rat spinal cord cells, determined by quantitative analysis of immunofluorescence punctate staining of Alexa Fluor 647 conjugated to TeNT. We also confirmed the mediator role of the Synaptic Vesicle Glycoprotein II (SV2) in TeNT endocytosis. The molecular docking assays to predict potential TeNT epitopes showed the binding of both antibodies to the HCR/T domain. A higher incidence was found between N1153 and W1297 when evaluating candidate residues for conformational epitope.

## 1. Introduction

The therapeutic use of monoclonal antibodies (mAbs) has grown steadily, increasing the number of conditions and pathologies that can benefit from their specific binding to the target [1]. In addition to treating complex diseases, mAbs shows promise in treating infectious diseases, such as tetanus. Tetanus is a severe infectious disease caused by the spore-forming bacterium *Clostridium tetani* [2]. The infection begins when *C. tetani* spores in the environment are introduced into wounds, minor abrasions, and umbilical stumps [3]. Inside the tissue, the anaerobic condition induces spore germination with Tetanus Neurotoxin (TeNT) production, which spreads through tissue spaces into the blood and lymph vessels and reaches the nervous system at the neuromuscular junctions [4,5,6]. TeNT undergoes retrograde axonal transport in spinal cord motor neurons until it reaches the nervous central system. It inhibits the neurotransmission release of GABA and glycine at the postsynaptic cleft of inhibitory interneurons [6]. As a consequence, the imbalance of synaptic cleft causes motor neuron hyperactivity, leading to severe muscle spasms, serious breathing difficulties, and death [7].

The toxin is synthesized as a single polypeptide with 1315 amino acids and a relative molecular mass of 150 kDa. Naturally, the toxin is found as a nontoxic precursor protein, which acquires its toxicity when cleaved into a light chain (LC) (50 kDa) (fragment A) and a heavy chain (HC) (100 kDa) [8]. The heavy chain comprises the N-terminal domain, HN, and the C-terminal domain, HCR/T, linked by a disulfide bond [5]. The LC is the catalytic domain responsible for vesicular synaptic membrane protein 2 (VAMP2) cleavage, resulting in the inactivation of inhibitory release of GABA and glycine. The HN domain is responsible for LC translocation to cytoplasm, and HCR/T is responsible for the molecular TeNT binding to the cell [4]. The HCR/T structure is composed of jelly-roll and β-trefoil domains. The TeNT entry into motor neurons requires the ganglioside GT1b receptor binding on the plasma membrane by the HCR/T. This domain presents two carbohydrate-binding sites: the lactose-binding site (the “W” pocket) and the sialic acid-binding site (the “R” pocket), present in the β-trefoil domain in HCR/T [9].

Globally, in the last ten years, more than 120,000 cases of tetanus have been reported, mainly in underdeveloped countries [10]. In the same period, 2590 cases were registered in Brazil, and the average lethality remains around 28% [11]. Tetanus prevention involves vaccination with the formaldehyde-inactivated tetanus toxoid (TT). Rapid treatment using immunoglobulins is recommended in accidents with potential transmission of tetanus, as they can neutralize circulating toxins. Available therapies are based on administering polyclonal antibodies produced in immunized horses or collected from human donors. This can lead to adverse effects due to the heterologous origin or risks from adventitious agents.

Previously, we sorted B cells from vaccinated individuals to develop better tetanus treatment. From a panel of mAbs screened by ELISA against tetanus toxoid, tetanus toxin, and fragment C (HCR/T), we found two different mAbs (TT-117 e TT-140) binding to fragment C that could also completely inhibit the binding of TeNT to GT1b. However, they did not neutralize the toxin when tested individually in vivo. The neutralization effect was achieved by combining three other mAbs [12,13]. As TeNT binds to the neurons via GT1b, and the two mAbs inhibit this binding when GT1b is immobilized in the plastic well of microplates, this study set out to understand the role of these two mAbs using cell-based assays and in silico approaches. Working with rat spinal cord neurons, we analyzed TeNT endocytosis, synaptic vesicle glycoprotein II (SV2) participation in the process, and the role played by these two mAbs.

In the evolution of monoclonal antibody selection techniques, in silico methods play a promising role in understanding the structure and interaction of mAbs with antigens. Characterizing paratopes and epitopes can provide important information in identifying the residues involved in the antigen-antibody interaction but does not provide specific details about the interactions of these residues. This aspect can be overcome through evaluation using docking methods. Molecular docking tools allow us to predict the binding interfaces between two proteins. Docking algorithms are typically divided into two steps: sampling and scoring. In the sampling stage, thousands of possible conformations are generated, while in the scoring stage, these conformations are classified to identify those that most closely match the native conformation [14]. Binding prediction by molecular docking methods was applied to mAbs TT-117 and TT-140.

## 2. Results

### 2.1. Tetanus Toxin Internalization Mediated by SV2 in RSpN Culture

First, we checked the viability of primary cell culture using the MTT method and found no significant alteration when adding TeNT, TT-117, TT-140, or isotype control (Figure 1).

The RSpN (neonatal rat spinal cord) culture (0.5 × 10^5^ cells/well) was incubated with TeNT-647 [40 nM] for 1 h on the seventh day of culture, allowing TeNT-647 into the intracellular space. We also labeled SV2 with an anti-SV2 antibody to evaluate its localization relationship with TeNT-647 and confirmed its mediating role in TeNT-647 endocytosis by the cellular model used (Figure 2A). The single experiment was performed in duplicate. The Pearson correlation coefficient measured the values of the control group (0.125), SV2 (0.275), and TeNT-647 (0.24), indicating a positive relationship between SV2 and TeNT-647 (0.655) (Figure 2B).

### 2.2. Tetanus Toxin Internalization Inhibition by mAbs

After confirming TeNT internalization in RSpN cells, we tested the effect of two antitetanus mAbs on toxin internalization when used individually in a single tested experimental condition (Figure 3A). Each mAb, TT-117 and TT-140 could separately and significantly inhibit TeNT entry into neuronal cells by 56.8% and 54.5%, respectively (Figure 3B). The data represented the mean ± SD of TT-117 (* *p* < 0.0127) and TT-140 (* *p* < 0.0266). We also analyzed the cell nucleus puncta to calculate the TeNT-647 puncta per cell (Figure 3C). The data represented the mean ± SD of positive control (** *p* < 0.0040), TT-117 (*** *p* < 0.0002), and TT-140 (** *p* < 0.0071). The data were obtained from two independent experiments in quadruplicate (eight wells). No statistical difference was observed between the two mAbs.

### 2.3. Molecular Docking

Six clusters were obtained for Molecular Docking performed in HADDOCK 2.4, each with four models for mAb TT-117, and ten clusters with four models each for mAb TT-140. For the first, eight models presented the binding region to the TeNT heavy chain, specifically to the HCR/T domain (Figure 4A). For the second, eight models presented the TeNT heavy chain binding region, four of which were directed to the HCR/T domain (Figure 4B). For the Molecular Docking of TT-117 carried out in SnugDock, 93 out of 98 models obtained presented their binding region to the HCR/T domain of TeNT. For mAb TT-140, 63 of the 80 models obtained presented their binding region to the HCR/T domain of TeNT. To demonstrate the binding region, the alignment of the ten models obtained from complex 1 of each antibody is shown in Figure 5A and Figure 5B, respectively. Table 1 presents the results scheme.

### 2.4. Epitopes Prediction

The epitopes of the complexes that bind to the HCR/T domain were determined. The mAb TT-117 presented 159 residues with hydrophobic interactions and 121 with hydrogen bonds. The mAb TT-140 showed 170 residues with hydrophobic interactions and 89 with hydrogen bonds. The residues were classified by incidence, with the highest incidence being close to 4% and an average of 1.86%. For this reason, residues with an incidence greater than 1.8% were determined as residues with a greater probability of composing the epitopes. Among all the residues with the highest incidence of composing the epitope, those with an incidence greater than or equal to 3% are included in the region between residues N1153 and K1297. These epitopes are shown in Table 2 and Table 3, respectively, for hydrophobic interactions and hydrogen bonds. Residues present in the β-trefoil domain are highlighted in red.

Figure 6 shows the molecular representation of the residues presented in Table 2 and Table 3. Tetanus toxin is represented in Figure 6A. Figure 6B–D illustrate a more detailed view of the hydrophobic interactions. The residues with the hydrogen bonds can be seen in Figure 6E–G. All these residues belong to the HCR/T domain.

## 3. Discussion

Different in vitro models have been explored to evaluate *Clostridium* toxins (CTNs) and endocytosis triggered by the binding of gangliosides on the surface of neuronal cells and motor neurons [4,15,16,17]. Gangliosides are predominantly found in neuronal cell membranes and are receptors for many biological toxins [18]. The GT1b exogenous treatment strategy in permanent cell lines was used for conditioning toxin internalization in neuronal cells Neuro-2a and PC 12 and the non-neuronal cells Hela [4]. Furthermore, efforts involving CTN inhibition assays with fully human or chimeric mAbs in Neuro-2a and PC12 cells have been previously described [19,20].

We first tried to work with permanent neuron cell lines without establishing a reliable model, probably because adherent cell lineages need detaching of the substrate for cell replication with agents that modify the cell’s membrane composition. Even when GT1b exogenous treatment was tried, we failed. We could only observe the TeNT entry into the cells when working with primary neuron cells. The initial experiments were crucial in finding a specific condition of cell density inoculation and incubation time that preserved cell viability and integrity (Figure 1). We could then proceed to characterize the endocytosis of TeNT-647, including the mediating role of SV2 in the endocytosis of TeNT-647 in spinal cord-derived neurons, confirming its colocalization in the intracellular compartment by the colocalization of the puncta number of TeNT-647 and SV2. This interaction has been previously reported in rat primary cultures, such as the hippocampus, which contains high ganglioside concentrations in lipid rafts [21].

To exert its toxic effects, TeNT needs to interact with neuronal cells at the muscle junction, retrogradely translocate to nerve cells and inhibitory neurons to cleave VAMP2, and inhibit the release of GABA and glycine neurotransmitters. The entry of TeNT into neuronal cells is mediated by the binding of the C-terminal region of the toxin heavy chain (HCR/T), or the so-called fragment C, to gangliosides present in neuronal cells, mainly GT1b. Previously, we obtained a panel of human anti-TeNT mAbs, of which two (TT-117 and TT-140) showed binding to fragment C and inhibition in the ELISA assay of TeNT binding to GT1b with an estimated IC_50_ of 2.4 × 10^−8^ M for both individual TT-117 and TT-140 [13]. Furthermore, Western blot analysis and peptide array described previously agree with fragment C as an epitope for the two mAbs at different residues [12]. However, when tested individually or mixed in vivo in a mouse model, they did not demonstrate protection at the concentrations used. In this work, we examined the entry and localization of TeNT in RSpN cells and the role of the two mAbs in TeNT internalization. We also tried to determine, by in silico approaches, the molecular interaction of the two mAbs with the TeNT.

Primary neuronal cells have been reported as an effective model for testing antitetanus mAbs, as they are highly susceptible to TeNT [22]. When rat primary cerebellar granule neurons (CGNs) were treated with a mixture of 50 nM fluorophore-labeled HCR/T and TT104-Fab at a 2:1 molar ratio, a partial inhibition of TeNT entry was reported. Furthermore, the binding of TeNT to GT1b was completely prevented by TT104-Fab in the ELISA [23]. In agreement, our results with another cellular composition, neonatal rat spinal cord (RSpN) cells, for evaluating the entry of TeNT was successful and confirmed the role of gangliosides in toxin internalization [24,25]. This environment comprises subpopulations of neuronal and non-neuronal cells (projection neurons, interneurons, motor neurons, meningeal fibroblasts, endothelial cells, and glia) [26,27]. Characterizing TeNT endocytosis and then evaluating its inhibitory process by mAbs in RSpN culture shows relevance involving the capacity for TeNT endocytosis in cell subpopulations. The adopted model established an effective platform for in vitro assays, as well as cerebellar and hippocampal primary cultures already described in the literature, due to the high pattern of expression of the ganglioside GT1b in cells of the central nervous system [17,21,23].

Our results, confirming previous literature data and establishing the model, led us to test the mediating role of mAbs TT-117 and TT-140 mAbs in TeNT entry into RSpN cells. Each antibody was able to inhibit the entry of TeNT-647 by percentages higher than 50% under the conditions tested; these were promising results due to their potential to protect people from developing tetanus conditions after accidents.

When analyzing the molecular docking results, we observed that for mAb TT-117, 30% of the models in the HADDOCK 2.4 docking and 93% of the models in the SnugDock docking presented the binding region in the HCR/T domain. For mAb TT-140, we observed that 10% and 63% showed a binding region in the HCR/T domain, respectively. Evaluating the residues shown in the binding region, we observed that most are part of the β-trefoil domain of HCR/T, where the W and R pockets, responsible for the binding to the GT1b receptor, are located. The region between residues N1153 and K1297 presents the highest incidence of residues that make up the epitope. Although they do not comprise the W pocket, composed of residues H1271, W 1289, and Y1290 [9], nor the R pocket, composed of residues D1147, N1216, D1214, R1226, and Y1229 [28], they are in a region close to these critical residues. Previous results indicated that the two mAbs bind to the HCR/T domain of the toxin but not to the W pocket region. Similarly, the docking results indicate an antibody interaction at the HCR/T domain but not directly at the residues within the W pocket.

Binding to the HCR/T domain, even if not to the W pocket, can alter the conformation of TeNT, inhibiting TeNT binding to gangliosides. Lukic et al. [29] described a mAb that binds to an epitope located on the TeNT light chain and is capable of completely inhibiting the binding of TeNT to the GT1b receptor, which can be explained by the steric hindrance of the HCR/T region of the antibody in binding the ganglioside. In a previous study, the same group described two neutralizing mAbs that bind to the HCR/T region, not in the β-trefoil domain, suggesting that the HCR/T may contribute to the neutralization of TeNT by steric hindrance in the β-trefoil domain towards its receptor. It was also seen by Yousefi et al. [30], who described six anti-HCR/T mAbs that bind to the ganglioside GT1b, with only one reacting to the β-trefoil domain when analyzed by peptide mapping. The inhibitory activity of the other mAbs can be explained through the steric hindrance of the HCR/T region in the GT1b ganglioside binding site or conformational changes in the GT1b binding domain of the toxin. In concordance with our results, Zhang et al. [31] described two mAbs that neutralize TeNT. Docking showed that none of them recognize the epitopes that make up the W pocket or the R pocket despite binding to the HCR/T domain, favoring the hypothesis that the neutralizing action may be due to a conformational change or steric hindrance. Steric hindrance may also explain the inhibition of mAbs TT-117 and TT-140 on TeNT-GT1b binding. The in silico results reported here confirm our previous peptide mapping array results [12]. Identifying epitopes on a large toxin such as TeNT is critical for the intended composition of mAbs [32].

The in vitro results, showing the inhibition of TeNT entry into neuronal cells and in silico, pointing out the residues involved in the interaction of mAbs with TeNT, reported here, encourage us to continue with the development of mAbs TT-117 and TT-140. We cannot run repetitions because of limitations on the number of neonatal rats we could use. Increasing the concentration of the mAbs could increase their effect. Other sequences clonally related to them will also be analyzed. An in vivo approach will also be tested with the two mAbs in different compositions. Preliminary results with TT-117 or TT-140 paired with another anti-TeNT mAb showed protection in mice. 

## 4. Materials and Methods

### 4.1. Tetanus Toxin Labeling

Alexa Fluor^®^647 was attached to the tetanus toxin ((Cat. no. 19050A2; List Biological—Campbell, CA, USA), referred to TeNT-647, according to the manufacturer’s protocol from Alexa Fluor^®^647 Labeling Kit (Cat. no. A30009; Thermo Fisher Scientific—Waltham, MA, USA) and stored at −20 °C. Quantifying the Alexa Fluor^®^647 ratio per tetanus toxin was obtained by scanning measurements using the NanoDrop One c device (Thermo Fisher Scientific—Waltham, MA, USA) with 280 and 650 nm readings. The average dye per mole of protein obtained was 1.7 moles.

### 4.2. Isolation and Cell Culture of Rat Spinal Cord Neurons (RSpN)

The primary culture of mixed neurons was obtained from the whole spinal cord of neonatal Wistar rats (Rattus norvegicus, 1–2 days old). The animals used in this study were approved by the Butantan Institute Animal Ethics Committee (CEUA no. 8850200323). With the assistance of dissection tools, the spinal cords were immersed in a Petri dish containing sterile ice-cold balanced salt solution (BSS) (pH 7.4, 120 mM NaCl, 5 mM KCl, 1.2 KH_2_PO_4_, 1.2 mM MgSO_4_·7H_2_O, 25 mM NaHCO_3_, and 13 mM Glucose). The spinal cords were washed three times with ice-cold BSS; the tissue was gently cut into pieces and digested with 200 µL of 2.5% Trypsin (Cat. no. 15090046; Gibco, Thermo Fisher Scientific—Waltham, MA, USA) at 37 °C for 30 min on an orbital shaker in 15 mL plastic tubes. Using a Pasteur pipette, the reaction was interrupted by homogenizing the tissue with 0.06% Trypsin Inhibitor (Cat. no. 17075029; Gibco, Thermo Fisher Scientific—Waltham, MA, USA). Cells were centrifuged (365× *g* for 15 min) resuspended in Neurobasal™-A Medium (Cat. no. 10888-022; Gibco, Thermo Fisher Scientific—Waltham, MA, USA) supplemented with 0.25 mM GlutaMAX™ (Cat. no. 35050061; Gibco, Thermo Fisher Scientific—Waltham, MA, USA), 2% B-27™ (50×), serum-free (Cat. no. 17504044; Gibco, Thermo Fisher Scientific—Waltham, MA, US) and 40 mg/L gentamicin (10 mg/mL) (Cat. no. 15710064; Gibco, Thermo Fisher Scientific—Waltham, MA, USA) and seeded between 0.5 × 10^5^–1 × 10^5^ per well in 96 well plate coated with 10 µg/mL Poly-D-lysine hydrobromide (Cat. no. P7886; Sigma-Aldrich—Rahway, NJ, USA). The medium culture was changed three hours after the seeding and every two days until the seventh day when the cells developed a typical morphology of a mixed neuron culture. Cell viability was quantified by the MTT method [33] at a final concentration of 0.5 mg/mL and incubated at 37 °C for 4 h. DMSO was added to solubilize formazan crystals produced by MTT reduction reaction for 15 min and its quantification was performed by Spectramax^®^—Molecular Devices using 570 nm optical density reading.

### 4.3. Tetanus Toxin Internalization Inhibition Assay

The TeNT entry inhibition by mAbs was initiated by the pre-incubation of 25 nM TeNT-Alexa-Fluor 647 and 25 nM mAbs in Neurobasal-A medium on an orbital shaker at 37 °C, 5% CO_2_ for 1 h. The mixtures were added to the cells for 1 h at 37 °C and 5% CO_2_. Cells were washed twice in 2% formaldehyde solution in PBS and processed for immunofluorescence.

### 4.4. Immunofluorescence

Cells were fixed in 4% paraformaldehyde in PBS for 30 min at room temperature, washed twice in PBS, and permeabilized with 0.5% (*vol*/*vol*) Triton X-100 in 4% (*vol*/*vol*) formaldehyde in PBS for 5 min at room temperature. The Hoechst dye (1:1000. Cat. no. H3570; Life Technologies, Thermo Fisher Scientific—Waltham, MA, USA) was used to stain DNA, and Alexa Fluor^®^ 488 Phalloidin (1:1000. Cat. no LA12379; Life Technologies, Thermo Fisher Scientific—Waltham, MA, USA) to stain the actin. Both dyes were used together and diluted in PBS, and they were added to cells for 1 h at room temperature. The images were captured using a Confocal Microscope Leica TCS SP8. Version 4.2.5 of CellProfiler™ (Broad Institute—Cambridge, MA, USA) was used to quantify the number of punctas and fluorescence intensity of the toxin and evaluate the correlation between the toxin and SV protein using Pearson’s coefficient.

### 4.5. Statistical Analysis

The statistical analyses were performed with GraphPad Prism Software (version 8.2.1. GraphPad Software—Boston, MA, USA). Statistical significance in the RSpN culture assays was calculated using a 2-tailed, unpaired *t*-test. Data are plotted in interleaved bars with mean SD. For the tetanus toxin internalization assay, each bar is representative of one independent experiment, composed of at least three repetitions for each experimental condition. The interleaved bars with mean SD of tetanus toxin internalization inhibition are representative of at least three independent experiments. The results were significant if the *p* value was lower than 0.05 (*, *p* < 0.05).

### 4.6. Analysis of Antibody Sequences

The sequences of the antibodies’ variable regions were analyzed using the abYsis platform [34], available at http://www.abysis.org/abysis/ (accessed on 7 July 2023), to identify the residues that make up the Complementarity-determining regions (CDRs). The amino acid sequences were submitted separately to the platform, light and heavy chain, and the residues were determined according to the IMGT numbering [35].

### 4.7. Obtaining the 3D Structure of Antibodies

Since the crystallized structure of the antibodies is not available, the 3D structure of the antibodies was obtained by homology. Since differences in variable domains, known as Fv, are responsible for these molecules’ structural and functional diversity, most structure predictions are based on Fv modeling. For Fv modeling, the ABodyBuilder platform [36], available online at the following address: https://opig.stats.ox.ac.uk/webapps/sabdab-sabpred/sabpred/abodybuilder2/ (accessed on 8 July 2023), was used. The amino acid sequences of the light and heavy variable chains were used as inputs to obtain the Fv domain by homology.

### 4.8. Molecular Docking by HADDOCK 2.4

To perform the docking on the HADDOCK 2.4 platform [37] (https://wenmr.science.uu.nl/haddock2.4/ (accessed on 9 July 2023)), the antibody structure was renumbered as a single chain without insertion nomenclature (number followed by a letter). The PyMOL Molecular Graphics System program (version 2.0 Schrödinger, LLC) was used for this. The PDB file of the structure obtained by homology was opened in the program, and on the command line, “renumber chain H” was written to renumber the heavy chain starting from residue 1, and then sat the command “renumber chain L, X,” where X is the residue number after the last residue of the heavy chain. The antigen’s PDB structure was also optimized before the Molecular Docking using the PDB tools website, available at https://wenmr.science.uu.nl/pdbtools/submit (accessed on 9 July 2023). The command “pdb_delheatm” was used to remove the water molecules, ions, and non-protein. As input files, it used the 3D structure of the antigen (code 7BY5), obtained from the PDB database (https://www.rcsb.org/ (accessed on 1 July 2023)), and the 3D structure of the antibody obtained by homology. For the input of molecule 1 (antibody), the active residues were flagged by inserting the number of residues that make up the CDRs previously defined (see the topic “Analysis of antibody sequences”). For this molecule, the option to automatically determine the residues as passive residues around the active residues was disabled. For the input of molecule 2 (antigen), the active residues were not flagged as there is no information about the epitopes, and the option to automatically define the surface residues as passive was enabled. In the sampling parameters, the number of structures in the rigid docking was defined as 10,000, 400 as the number of structures in the semi-flexible refinement, 400 as the number of structures in the final refinement, and 400 as the number of structures to be analyzed. The clusters obtained are classified according to the FCC (Fraction of Common Contacts) criterion. Generally, solutions with an FCC value greater than 0.6, considered a good value of structural complementarity, are selected. This criterion compares the density map of the protein-protein complex with the density map of each protein individually.

### 4.9. Molecular Docking by SnugDock

To perform the Docking using the SnugDock [38] program on the ROSIE platform (https://rosie.rosettacommons.org/ (accessed on 9 July 2023)), the input must be a single PDB with the antibody-antigen complex in an initial conformation considered close to native. The input PDB must be ordered with the light chain (L) coming first, then the heavy chain (H), and finally the antigen chain (A). For this, the complex was formed on the Zdock [39] online platform (rigid docking), available at https://zdock.umassmed.edu/ (accessed on 9 July 2023). To submit the docking in Zdock, the PDB of the pre-treated toxin was attached to ligand 1 (rigid structure), and ligand 2, the PDB of the homology of each mAb was attached, with each chain renumbered from residue 1 to mitigate possible errors due to insertions (movable structure). Ten complexes were obtained in Zdock for each antibody, and these were reordered so that the chains were in the LH_A sequence for submission in SnugDock. From each complex submitted at SnugDock, 1000 models were obtained. The complexes obtained were summarized into 10 best models with the lowest conformational energy.

### 4.10. Determination of Epitopes

We used the LigPlot+ platform [40] to determine the antigen (epitopes) binding residues in the antigen-antibody complex obtained by docking. The diagrams depict the interaction patterns of hydrogen bonds and hydrophobic contacts. We used the PDB file of the antigen-antibody complex obtained by docking as an input file. Since the mAbs structures for the HADDOCK platform were renumbered as a single chain, the complex obtained was composed of two chains, one for the mAb and one for the antigen, which remains to use the DIMPLOT program at the LigPlot+ platform, which predicts interactions between a protein-protein interface. As for the SnugDock complexes, can be used in the ANTIBODY program at the LigPlot+ platform since the mAb heavy and light chain was identified separately. At the program, the mAb heavy chain is typed as H, the mAb light chain is typed as L, and the antigen chain is typed as A and for the topic “Antibody Loop Numbering Scheme” was typed “Other” since the IMGT numbering system was used (not available to be select at the LigPlot+).

## Figures and Tables

**Figure 1 ijms-25-05788-f001:**
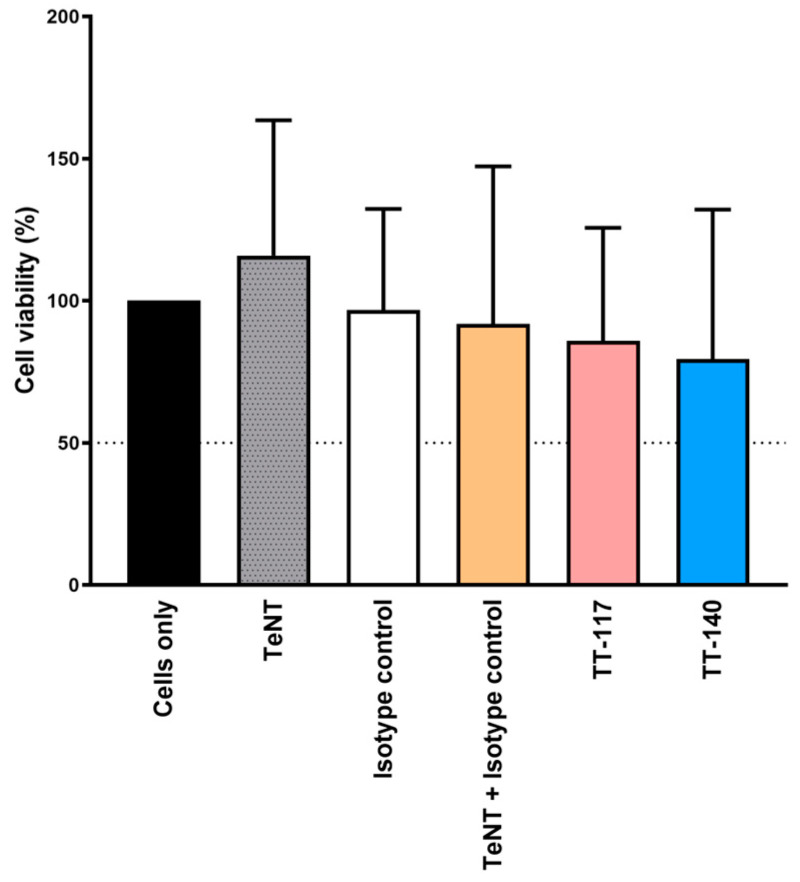
Cell viability percentage measured by MTT assay. RSpN cells (0.75 × 10^5^ cells/mL) were exposed to TeNT and mAbs. The TeNT was pre-incubated with antitetanus mAbs or isotype control for 1 h at 37 °C (1:2 molar ratio, 25 mM). The experimental conditions comprised control (cells only in the black bar), TeNT in the gray bar, isotype control in the white bar, TeNT plus isotype control in the orange bar, TT-117 mAb in the pink bar, and TT-140 in the blue bar. The percentages are the average of two independent experiments in quadruplicate each.

**Figure 2 ijms-25-05788-f002:**
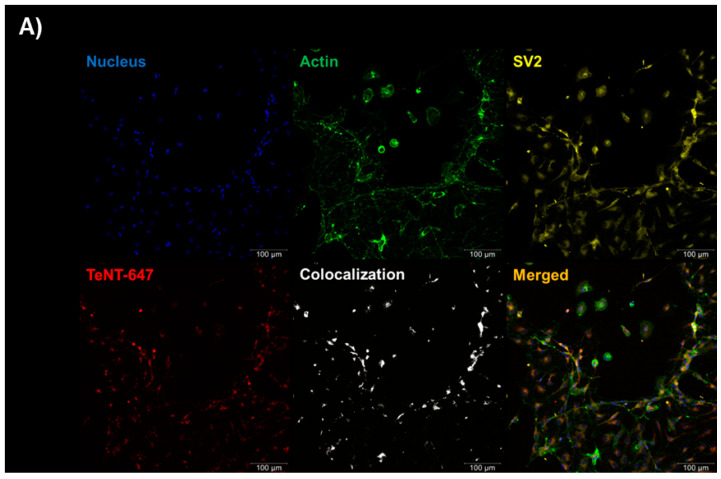
SV2 mediator role and tetanus toxin intracellular presence in RSpN. Cells were incubated with 40 nM TeNT-647 in the culture medium for 1 h at 37 °C and fixed for immunofluorescence. (**A**) Immunofluorescence staining for nucleus (blue), actin (green), synaptic vesicle protein 2 (yellow), and TeNT conjugated with Alexa Fluor 647 dye [40 nM] (red). The association between TeNT-647 and SV2 is represented in white. The merged images show the TeNT internalized by neuron. (**B**) Pearson’s correlation coefficient quantified the SV2 stain (purple) and TeNT-647 (blue) puncta’s number and associated their colocalization in the intracellular compartment (gray bar). Control group, cells only is the black bar. The value in the graph is averaged from three images, each one in one well, taken from one independent six-well experiment. Statistical analysis was performed using the unpaired *t*-test (GraphPad Prism 8.2.1). *p* values are reported or indicated by * and ** for *p* < 0.05 and *p* < 0.01, respectively. Immunofluorescence images at 20× magnification and scale bars = 100 μm.

**Figure 3 ijms-25-05788-f003:**
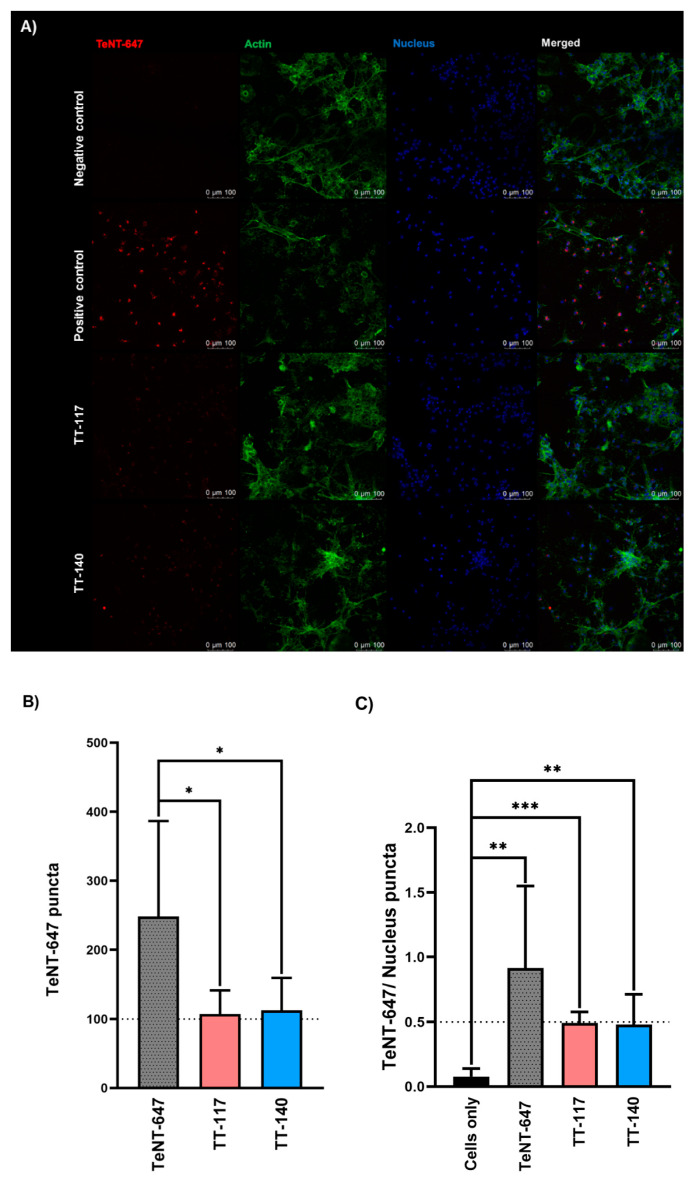
TT-117 and TT-140 mAbs prevent TeNT-647 internalization in RSpN by interfering with TeNT HCR/T binding to receptor GT1b. The teNT-647 and mAbs mixture was pre-incubated for 1 h (1:1 molar ratio, 25 nM). (**A**) Experimental conditions comprised negative control (RSpN cells only), positive control (RSpN cells incubated with TeNT-647), and cells incubated with the toxin-antibody mixture. Immunofluorescence staining for TeNT-647 (red), actin (green), nucleus (blue), and the merged image. Immunofluorescence images at 20× magnification and scale bars = 75 µm. (**B**) Quantified puncta number experimental condition: no treatment (gray), TT-117 (pink), and TT-140 (blue). (**C**) The ratio between TeNT-647 and cell nucleus puncta number comprised negative control (cells only (black)), TeNT-647 only (grey), and cells incubated with the toxin-antibody mixture (pink for TeNT and TT-117 and blue for TeNT and TT-140). TeNT-647 and cell nucleus puncta were quantified by CellProfiler software version 4.2.5. The value in the graph is averaged from two independent experiments in quadruplicate each. Statistical analysis was performed using the unpaired *t*-test (GraphPad Prism 8.2.1). *p* values are reported or indicated by *, **, and *** for *p* < 0.05, *p* < 0.01, and *p* < 0.001, respectively.

**Figure 4 ijms-25-05788-f004:**
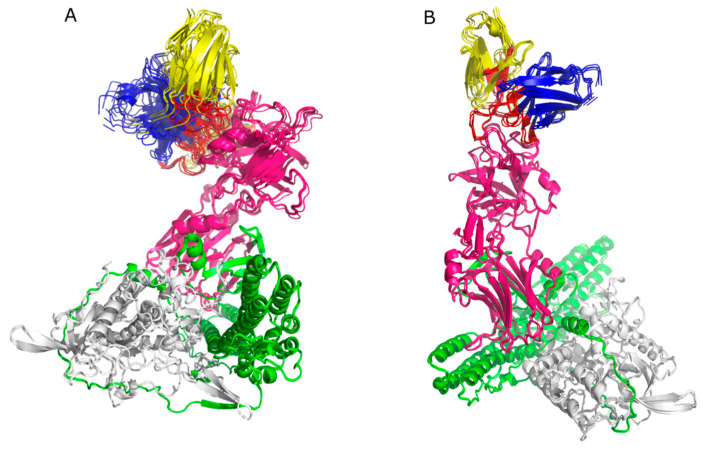
HADDOCK 2.4 docking. The alignment for the TT-117’s eight models (**A**) and the alignment for the TT-140’s four models (**B**). The light chain of TeNT is colored gray, the HN domain is green, and the HCR/T domain is pink. The variable HC of the mAbs is shown in blue and the variable LC of the mAbs is yellow. CDRs are highlighted in red.

**Figure 5 ijms-25-05788-f005:**
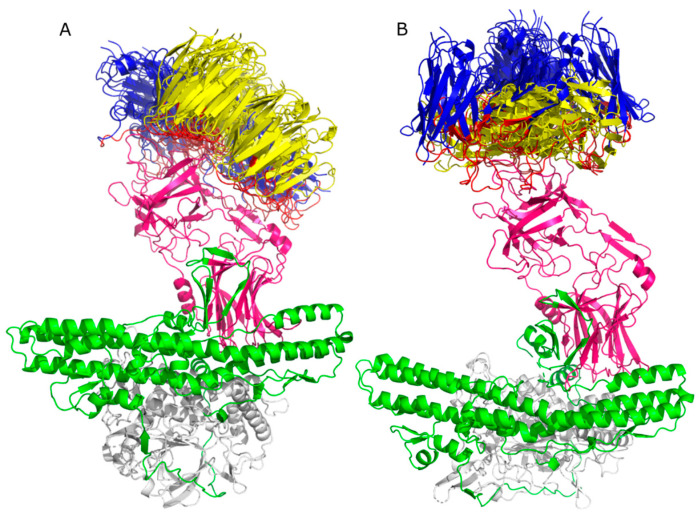
SnugDock docking. Alignment for ten models of the TT-117 cluster 8 (**A**) and alignment for ten models of the TT-140 cluster 6 (**B**). The TeNT light chain is white, the HN domain is green, and the HCR/T domain is pink. The variable HC of mAbs is shown in blue, and the variable LC of the mAbs is shown in yellow. The CDRs are highlighted in red.

**Figure 6 ijms-25-05788-f006:**
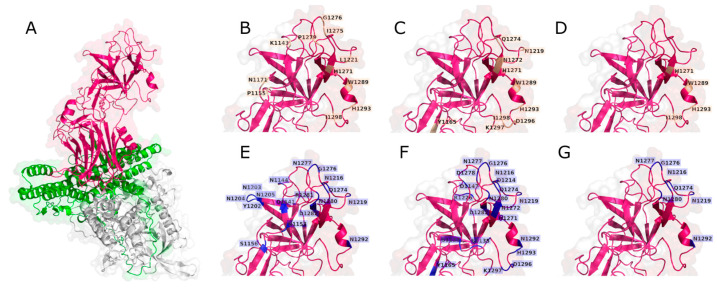
PDB structure of tetanus toxin-TeNT (**A**) with the residues comprising the hydrophobic interactions for mAb TT-117 (**B**) and mAb TT-140 (**C**), and the hydrogen bond interaction residues for mAb TT-117 (**E**) and TT-140 (**F**). (**D**,**G**) show the residues common for both mAbs for hydrophobic interactions and hydrogen bonds, respectively. The TeNT light chain is white, the HN domain is green, and the HCR/T domain is pink. The hydrophobic interactions are shown in purple and hydrogen bonds are shown in blue.

**Table 1 ijms-25-05788-t001:** Results obtained for the Molecular Docking.

mAb	Program	Input	Output	Clusters	Total Models	Models Binding HCR/T
TT-117	HADDOCK 2.4	1 *	400	6	24	8
TT-140	HADDOCK 2.4	1 *	400	10	40	4
TT-117	Snug Dock	10 **	10,000 ***	-	100	93
TT-140	Snug Dock	10 **	10,000 ***	-	100	63

* The input consists of the mAb and antigen sequence. ** The input consists of the 10 complexes obtained in the Zdock docking. *** For each complex submitted, 1000 complexes are obtained.

**Table 2 ijms-25-05788-t002:** Results obtained for the epitopes prediction—hydrophobic interactions.

TT-117		TT-140	
Residue	Incidence	Residue	Incidence
HIS 1293	2.56%	TRP 1289	3.06%
HIS 1271	2.41%	HIS 1271	3.06%
ASN 1171	2.34%	ASN 1219	2.64%
ILE 1275	2.11%	HIS 1293	2.64%
ILE 1298	2.11%	LYS 1297	2.53%
LEU 1221	1.96%	ASN 1272	2.53%
GLY 1276	1.96%	ASP 1296	2.53%
PRO 1279	1.89%	ILE 1298	2.53%
PRO 1155	1.89%	TYR 1165	2.11%
LYS 1143	1.89%	GLN 1274	1.90%
TRP 1289	1.81%		

**Table 3 ijms-25-05788-t003:** Results obtained for the epitopes prediction—hydrogen bonds.

TT-117		TT-140	
Residue	Incidence	Residue	Incidence
ASN 1219	3.92%	HIS 1271	4.27%
ASN 1277	3.92%	ASN 1219	3.96%
ASN 1153	3.23%	ASN 1280	3.66%
ASN 1216	3.23%	LYS 1297	3.66%
GLN 1274	3.23%	ASN 1277	3.35%
ASN 1203	3.00%	ASP 1296	3.35%
ASN 1205	3.00%	ASN 1272	3.05%
ASN 1280	2.76%	ASN 1216	3.05%
SER 1156	2.30%	GLN 1274	3.05%
ASN 1204	2.07%	ASP 1278	2.44%
ASN 1292	2.07%	ASP 1282	2.44%
ASN 1144	1.84%	HIS 1293	2.44%
ARG 1281	1.84%	SER 1135	2.44%
GLN 1141	1.84%	ASP 1214	2.13%
GLY 1276	1.84%	ARG 1226	2.13%
TYR 1202	1.84%	ASN 1163	2.13%
ASP 1282	1.84%	ASN 1292	2.13%
		ASP 1147	1.83%
		SER 1137	1.83%
		GLY 1276	1.83%
		TYR 1165	1.83%

## Data Availability

The variable sequences of the mAbs were deposited at GeneBank and received accession numbers from OP373409 to OP373418 related to the heavy and light chains of mAbs TT-117 and TT-140. A preprint of the manuscript is available at bioRxiv.org (https://doi.org/10.1101/2024.02.25.582016).

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
