# Peer review of "A Dual Strategy—In Vitro and In Silico—To Evaluate Human Antitetanus mAbs Addressing Their Potential Protective Action on TeNT Endocytosis in Primary Rat Neuronal Cells"

_ijms, 2024, doi:10.3390/ijms25115788_

Round 1

Reviewer 1 Report

Comments and Suggestions for Authors

The authors investigated the protective effects of anti-tetanus mAbs on TeNT endocytosis via a dual strategy in neuronal cells. The subject is interesting for the clinic, and the manuscript is well-written. However, some small parts need correction and improvement.

-          Why did the authors present the results of immunofluorescence staining in 2D culture in Figure 1 one-independently? The results of at least three independent experiments should be included.

-          Authors only need to include the relevant P values in the figures' legends. For example, if the graph does not have ***, it should not be in the description.

-          The scale bar on the picture is hard to read in Figure 2. Scala bar length and magnification information should be added to the figure legends of microscope images.

-          Cell viability results should be included in the study.

-          What is the ethical approval date? How many animals were used for primary culture experiments? Why did the authors not conduct and present in vivo animal experimental research in this study? Why was ethical permission obtained for only primary culture formation?

-          The authors did not include some previous reports on monoclonal antibodies to tetanus toxin in the discussion part. Two examples are given below. The authors should discuss and compare their results with similar studies in a separate paragraph.

o    Li, Y., Chen, Y., Cui, J., Liu, D., Zhang, W., Xue, C., Xiong, X., Liu, G., & Chen, H. (2023). Preparation and characterization of a neutralizing murine monoclonal antibody against tetanus toxin. Journal of immunological methods, 513, 113427. https://doi.org/10.1016/j.jim.2023.113427

o    Zhang, G., Yu, R., Chi, X., Chen, Z., Hao, M., Du, P., Fan, P., Liu, Y., Dong, Y., Fang, T., Chen, Y., Song, X., Liu, S., Li, J., Yu, C., & Chen, W. (2021). Tetanus vaccine-induced human neutralizing antibodies provide full protection against neurotoxin challenge in mice. International immunopharmacology, 91, 107297. https://doi.org/10.1016/j.intimp.2020.107297

-          The authors mentioned the study's shortcomings but did not sufficiently highlight them. A separate paragraph on limitations should be added at the end.

Author Response

The authors investigated the protective effects of anti-tetanus mAbs on TeNT endocytosis via a dual strategy in neuronal cells. The subject is interesting for the clinic, and the manuscript is well-written. However, some small parts need correction and improvement.

We appreciate your time and consideration in reviewing our article. 

-          Why did the authors present the results of immunofluorescence staining in 2D culture in Figure 1 one-independently? The results of at least three independent experiments should be included.

We presented the results from one experiment. The number of neonatal mice we can use is restricted, but the result is clear. For your consideration, we add two additional panels with images from other wells. Each panel shown below was taken from a different well (there were six total). We changed Figure 2A in the manuscript to a more representative panel. The graph (Fig 2B) is the average of these three panels.

Panel 1:  

Panel 2:

Panel 3:

-          Authors only need to include the relevant P values in the figures' legends. For example, if the graph does not have ***, it should not be in the description.

That is correct and has been changed.

-          The scale bar on the picture is hard to read in Figure 2. Scala bar length and magnification information should be added to the figure legends of microscope images.

The scale bars were improved.

-          Cell viability results should be included in the study.

A figure with cell viability measured by MTT method was included in the manuscript.

-          What is the ethical approval date? How many animals were used for primary culture experiments? Why did the authors not conduct and present in vivo animal experimental research in this study? Why was ethical permission obtained for only primary culture formation?

Date of ethical approval: July 19th, 2023.

We used 15 to 20 animals to obtain cells to distribute to around 30 wells of a 96w microplate at 0,5 × 105 cells/well.

Ethical permission was granted for mice also. We did not include in vivo experiments as only one preliminary experiment was done, and we do not have control over when we can proceed with the in vivo experiments, as they are done by our Institution’s Biological Quality Control team when a vacancy is possible in their committed schedule with the release of Butantan products batches. 

-          The authors did not include some previous reports on monoclonal antibodies to tetanus toxin in the discussion part. Two examples are given below. The authors should discuss and compare their results with similar studies in a separate paragraph.

o    Li, Y., Chen, Y., Cui, J., Liu, D., Zhang, W., Xue, C., Xiong, X., Liu, G., & Chen, H. (2023). Preparation and characterization of a neutralizing murine monoclonal antibody against tetanus toxin. Journal of immunological methods, 513, 113427. https://doi.org/10.1016/j.jim.2023.113427

o    Zhang, G., Yu, R., Chi, X., Chen, Z., Hao, M., Du, P., Fan, P., Liu, Y., Dong, Y., Fang, T., Chen, Y., Song, X., Liu, S., Li, J., Yu, C., & Chen, W. (2021). Tetanus vaccine-induced human neutralizing antibodies provide full protection against neurotoxin challenge in mice. International immunopharmacology, 91, 107297. https://doi.org/10.1016/j.intimp.2020.107297

 The suggested reports were included in the article.

-          The authors mentioned the study's shortcomings but did not sufficiently highlight them. A separate paragraph on limitations should be added at the end.

The last paragraph now includes a sentence about the limitations on the number of neonatal rats. Also, there is a comment about the concentration of mAbs that was used. 

Reviewer 2 Report

Comments and Suggestions for Authors

Lima et al., found two anti-tetanus antibodies that inhibit uptake of the tetanus toxin into neurons. The authors also performed in silico analysis to determine the residue related to the interaction between antibodies and tetanus toxin. Their results were clear and showed the effectiveness of their antibodies to prevent the internalization in neurons. However, the manuscript needs few correction and clarification before publication.

In the abstract, authors described "A higher 27 incidence was found between N1153 and W1289". However, there are no explanation about these 2 residues in Result text. Please explain how you define the incidence of 2 residues is high because other residues, foe example N1219 and N1277 in TT-117, also shows high incidence. 

In figure 1A, is the image of the association between TeNT-647 and SV2 represented in white (v) merged image of iii and iv?

If so, the images do not appear to reflect the fluorescence intensity of images iii and iv, and it appears that the fluorescence intensity was altered when the images were merged. If the co-localization of the two proteins is to be shown, it would be easier to understand if the images were merged without changing the color, as in image vi.

In Figure 2B, how many cells were counted to calculate the number of TeNT-647 puncta? The number is affected by the counted number of cells and cell area. Authors should calculate the number of puncta per cell or cell area.  

Minor point

Page 2, Lane 93, "0.5x105 cells/well" should be "0.5x105 cells/well". Other indices need to be corrected as well.

For all images obtained by microscope, please show scale bars clearly.

In 4.2. section, KH2PO4, MgSO47H2O, and NaHCO3 should be corrected.

in Figure 1A, since there are multiple proteins that compose the cytoskeleton, it is more accurate to refer to them as actin. 

In Figure 1B, what is the Cell control? Please clarify the term.

Author Response

Lima et al., found two anti-tetanus antibodies that inhibit uptake of the tetanus toxin into neurons. The authors also performed in silico analysis to determine the residue related to the interaction between antibodies and tetanus toxin. Their results were clear and showed the effectiveness of their antibodies to prevent the internalization in neurons. However, the manuscript needs few correction and clarification before publication.

We appreciate your time and consideration in reviewing our article. Thank you for the suggestions. We have made improvements to the manuscript, with additional images, graphs, and considerations.

In the abstract, authors described "A higher 27 incidence was found between N1153 and W1289". However, there are no explanation about these 2 residues in Result text. Please explain how you define the incidence of 2 residues is high because other residues, foe example N1219 and N1277 in TT-117, also shows high incidence. 

W1289 was incorrectly typed. It should be K1297. It was changed in the manuscript.

The sentence “Among all the residues with the highest incidence of composing the epitope, those with an incidence greater than or equal to 3% are included in the region between residues N1153 and K1297” was included in Session 4.2. The abstract was corrected, and in the Discussion, the sentence:

Despite the location of the residues in the β-trefoil, they do not comprise the W pocket, which is made up of residues H1271, W1289, and Y1290 [8] nor the R pocket, which is made up of residues D1147, N1216, D1214, R1226 and Y1229 [27]

was changed to:

The region between residues N1153 and K1297 presents the highest incidence of residues that make up the epitope. Although they do not comprise the W pocket, which is composed of residues H1271, W1289, and Y1290 [8], nor the R pocket, which is composed of residues D1147, N1216, D1214, R1226, and Y1229 [27], they are in a region quite close to these important pockets.

In figure 1A, is the image of the association between TeNT-647 and SV2 represented in white (v) merged image of iii and iv?

If so, the images do not appear to reflect the fluorescence intensity of images iii and iv, and it appears that the fluorescence intensity was altered when the images were merged. If the co-localization of the two proteins is to be shown, it would be easier to understand if the images were merged without changing the color, as in image vi.

In Figure 1A (now 2A), the representation of the association between TeNT-647 and SV2 is shown in the merged image (labeled as "v"), which is indeed the merge with images iii and iv, shown in white color under the colocalization title. When the colocalization map (in white) is merged with the other channels, the software automatically adjusts the intensity. To avoid this problem of suppressing fluorescence intensity brightness, we structured the panel of Figure 2A to show the colocalization map in a separate image.

In Figure 2B, how many cells were counted to calculate the number of TeNT-647 puncta? The number is affected by the counted number of cells and cell area. Authors should calculate the number of puncta per cell or cell area.  

We included a graph (Fig 3C) in the manuscript to assess this point.

Minor point

Page 2, Lane 93, "0.5x105 cells/well" should be "0.5x105 cells/well". Other indices need to be corrected as well.

They were checked and corrected.

For all images obtained by microscope, please show scale bars clearly.

They were improved.

In 4.2. section, KH2PO4, MgSO47H2O, and NaHCO3 should be corrected.

Done

in Figure 1A, since there are multiple proteins that compose the cytoskeleton, it is more accurate to refer to them as actin. 

Thank you for addressing the point. It was changed. 

In Figure 1B, what is the Cell control? Please clarify the term.

Cells only. The legend was corrected.

Reviewer 3 Report

Comments and Suggestions for Authors

This work presents the experimental validation of human monoclonal antibodies (mAbs), which could inhibit TeNT endocytosis through binding the HCR/T domain of TeNT (tetanus toxin). This work can be encouraged but lack full evidences to support their conclusions. This work can be further refined according to the following points.

1. In part 2.1, although the colocalization percentage of SV2 and TeNT was high, what is the cell uptake perventage of TeNT? The flow cytometry assay is necessary to quantify the positive cell percentage. Small compound inhibitors or siRNA targeting SV2 should be used as the control to validate the mediator role of SV2.

2. In part 2.2, what is the IC50 of each mAb (TT-117 and TT-140)? The dosage- and time- dependent incubation experiment is needed.

3. In part 4.3, the author refer to “The mixtures and increased concentrations of TeNT- 297 Alexa-Fluor 647 (25 nM, 50 nM, and 100 nM) were added to cells”, no such results were presented.

4. The figure 1 should be labeled with the sample name in each treatment group.

5. The affinity constant KD between mAb and TeNT should be measured to validate the prediction.

6. The title should be short.

Author Response

This work presents the experimental validation of human monoclonal antibodies (mAbs), which could inhibit TeNT endocytosis through binding the HCR/T domain of TeNT (tetanus toxin). This work can be encouraged but lack full evidences to support their conclusions. This work can be further refined according to the following points.

We appreciate your time and consideration in reviewing our article. We expect that the improvements made are satisfactory. 

  1. In part 2.1, although the colocalization percentage of SV2 and TeNT was high, what is the cell uptake perventage of TeNT? The flow cytometry assay is necessary to quantify the positive cell percentage. Small compound inhibitors or siRNA targeting SV2 should be used as the control to validate the mediator role of SV2.

A figure (3C) was included for this consideration. We do not have siRNA targeting SV2.

We changed Figure 2A to be more representative (cells are most spread), and Figure 2B now represents the average of three panels with images taken from three separate wells (the experiment was repeated in six wells in the microplate).

See the three panels below:

Panel 1: 

Panel 2:

Panel 3:

  1. In part 2.2, what is the IC50of each mAb (TT-117 and TT-140)? The dosage- and time- dependent incubation experiment is needed.

These data were obtained and published previously. The reference is mentioned in the manuscript, and it was included in the manuscript.

The estimated IC50 was 2.4 × 10-8 M for both individual TT-117 and TT-140.

Manieri TM, Takata DY, Targino RC, Quintilio W, Batalha-Carvalho JV, da Silva CML, et al. Characterization of Neutralizing Human Anti-Tetanus Monoclonal Antibodies Produced by Stable Cell Lines. Pharmaceutics. 2022;14(10).

  1. In part 4.3, the author refer to “The mixtures and increased concentrations of TeNT- 297 Alexa-Fluor 647 (25 nM, 50 nM, and 100 nM) were added to cells”, no such results were presented.

This information is not correct and was changed. It was taken from a previous experiment with neuronal cell lineage (not included in the manuscript).

  1. The figure 1 should be labeled with the sample name in each treatment group.

The figure (now figure2) was changed.

  1. The affinity constant KD between mAb and TeNT should be measured to validate the prediction.

The affinity constant KD between mAb and TeNT was previously measured and published (Manieri et al, 2022), but the prediction made through molecular docking is unrelated to kD. Molecular docking assays can predict the interaction residues between two proteins but not the interaction strength between the two molecules.

mAb

Kinetic affinity

Steady State

Ka (M-1.s-1)

Kd (s-1)

KD (M)

KD (M)

TT-117

6.33 × 103

ND

ND

2.58 × 10−7

TT-140

8.65 × 104

3.92 × 10−5

4.54 × 10−10

1.10 × 10−7

ND = not determined. There was no dissociation in the period of the 36,000s.

Manieri TM, Takata DY, Targino RC, Quintilio W, Batalha-Carvalho JV, da Silva CML, et al. Characterization of Neutralizing Human Anti-Tetanus Monoclonal Antibodies Produced by Stable Cell Lines. Pharmaceutics. 2022;14(10).

  1. The title should be short.

We prefer to maintain the title. The journal has manuscript titles the same length as ours.

Round 2

Reviewer 3 Report

Comments and Suggestions for Authors

The authors have modified this manuscript to their best. It can be published in its present form.